# Effect of Slaughter Age on Muscle Fiber Composition, Intramuscular Connective Tissue, and Tenderness of Goat Meat during Post-Mortem Time

**DOI:** 10.3390/foods8110571

**Published:** 2019-11-13

**Authors:** Allah Bakhsh, Young-Hwa Hwang, Seon-Tea Joo

**Affiliations:** 1Division of Applied Life Science (BK21+), Gyeongsang National University, Jinju 52852, Korea; drallahbakhsh1@yahoo.com; 2Institute of Agriculture & Life Science, Gyeongsang National University, Jinju 52852, Korea; philoria@hanmail.net

**Keywords:** goat meat, goat age, muscle fiber composition, goat meat storage, goat meat quality, meat tenderness

## Abstract

This study evaluated the effects of slaughter age and post-mortem time on meat quality traits, tenderness, histochemical analyses, and perimysium thickness in the *longissimus thoracis* (LT) muscle of the Korean native black goat (KNBG) maintained at 4 °C for up to 21 days post mortem. Samples of LT muscle were obtained from the carcasses of 24 KNBGs, including old and young goats (AG, *n* = 12, 18 months of age; YG, *n* = 12, 9 months of age), to measure all analyses during 21 days of post-mortem time. AGs had a higher percentage of type I fiber but a lower percentage of type IIA fiber than YGs (*p* < 0.05). AGs had higher a* value, lower released water (RW) %, and higher Warner–Bratzler shear force (WBSF) value than YGs (*p* < 0.05). The perimysium thickness (PMT) of AGs was also higher than that of YGs (*p* < 0.05). Although the PMT did not change during post-mortem period, the WBSF value of AGs was higher than that of YGs after 21 days post mortem (*p* < 0.05). The results imply that AGs are tougher than YGs due to their muscle fiber characteristics and thicker perimysium.

## 1. Introduction

Goat meat is a good source of animal protein for many Asian countries, including Korea. The Korean native black goat (KNBG) has been raised as a domestic stock in Korea for approximately 2000 years. The KNBG is the only breed of goat native to Korea. About 80% of the population is predominantly black. The recent statistics in year 2018 show that 8028 farm households have raised 581,026 head of goats in Korea [1]. KNBG meat consists of low cholesterol and fat content, and a higher content of calcium and iron compared to beef and pork [2]. Furthermore, KNBG meat has a favorable effect on human health since it helps to control blood cholesterol levels [3]. Moreover, the goat is a good source of lean meat with desirable fatty acids, and it deposits a relatively higher proportion of polyunsaturated fatty acids compared to other ruminants [4]. However, the consumption of KNBG meat has not increased gradually like other meats because of its poor meat quality. Although several studies have been conducted to improve nutritional values [5], fatty acid profile [6], and slaughter method [7] for the KNBG, there are few studies on goat meat tenderness. Tenderness is one of the most important eating quality characteristics, as it defines consumer acceptance for meat [8]. This quality mostly depends on the properties of the following two main structural constituents of muscle tissue: myofibers and intramuscular connective tissue, and the interaction between these two structural factors [9]. After animal slaughter and subsequent rigor development, myofibers undergo a proteolytic breakdown (called the aging process) that causes the progressive tenderization of meat [9].

Meat quality can be affected by several factors, including breed, sex, age, slaughter weight, diet, management, slaughter procedure, and aging time [10]. The aging of meat refers to the act of holding meat post-rigor at cooling temperatures for a given period in order to improve meat traits quality specifically tenderness [11]. The meat of young or light-weight animals is preferred by consumers because it is a common perception that younger or light-weighted animals produce more tender meat than older or heavier animals [12]. The impact of animal age on meat quality traits is of particular importance, as it can affect marketing decisions—old animals are heavier and their age affects the tenderness and color of meat [13]. The intramuscular connective tissue sustains the integrity of the skeletal muscle and consists of three strata: namely the epimysium, which covers the whole muscle; the perimysium, which outlines the fascicles; and the endomysium, which encases the individual myofibers in a muscle. Among these three types, the perimysium is the most important because of its quantity (about 90%), and its pervasive organization is believed to be the main factor influencing the toughness of meat [14]. A comparison between the impact of post-mortem storage time and slaughter age on goat *longissimus thoracis* (LT) muscle can provide useful information.

Previously, numerous studies have been carried out on post-mortem storage time on porcine [15], bovine [16], and ovine [17] muscles, but existing literature on goat meat in relation to age and storage time is very limited. We hypothesize that goat meat might respond differently to post-mortem storage time at different slaughter ages. Therefore, the objective of the current study was to evaluate the effect of post-mortem time on muscle fiber composition, meat quality traits, and intramuscular connective tissues of goat loin muscle at various slaughter ages.

## 2. Materials and Methods

### 2.1. Animals and Samples

The study was approved and conducted in accordance with the guidelines of the Institutional Animal Care and Use Committee (IACUC) of the Division of Applied Life Science, Gyeongsang National University, South Korea (Approval ID: 125). Twenty-four castrated male KNBGs were selected based on age, including adult goats weighing 30.93 ± 0.5 kg (*n* = 12, 18 months of age, labeled as AG) and young goats weighing 14.33 ± 0.5 kg (*n* = 12, 9 months of age, labeled as YG) for the experimental trial. The experimental animals were raised on the Gyeongsang National University animal breeding farm, located at a distance eight miles from the main campus. The experimental animals were provided with rice straw and concentrate, including: corn 14.2%, wheat 15.0%, wheat bran 19.3%, tapioca 5%, corn gluten feed 16.0%, coconut meal 7.0%, distillers’ grains 5%, palm kernel grains 9.5%, canola meal 7.0%, molasses 3.0%, limestone 1.1%, and premix 1.9%. Feed was provided ad libitum twice daily. All experimental animals were allowed free access to fresh water placed in buckets. The transportation of experimental animals was carried out from the farm to the slaughterhouse one day prior to slaughter. The animals were rested for 18 h with free access to water, but not food. The slaughtering procedure was carried out in a slaughterhouse located in the vicinity of Gyeongsang National University Korea. Approximately 24 h after slaughter, the LT muscles were excised from the right and left sides of carcasses. The experiment used factorial design, with age and aging time as factors. A total of 24 LT muscles (12 muscles from each side) were randomly assigned for each aging day (0, 7, 14, and 21 days, *n* = 6). Following this, steaks of 5 cm thickness were prepared, vacuum packed, and stored at 4 °C. Muscle samples of about 20 g were used for histochemical analysis. The sample was taken from the center of the LT muscle within 30 min of slaughter and then frozen in liquid nitrogen.

### 2.2. Histochemical Analyses

From the entire block of frozen meat sample (1.0 × 1.0 × 1.5 cm^3^), transverse serial sections (10-μm thickness) were made with a cryostat microtome (HM525, micromesh, Walldrof, Germany) at −27 °C. Incubation was then carried out for histochemical analysis of myosin and adenosine triphosphatase (mATPase) followed by alkaline (pH 10.70) and acid (pH 4.63) pre-incubation using the method proposed by Brooke and Kaiser [18] with slight modifications. Stained sections were analyzed using an image analysis system (Image-Pro^®^plus 5.1, Media Cybernetics Inc., Rockville, MD, USA). Muscle fiber nomenclature was based on Brooke and Kaiser [18]. Fibers were divided into three types: I, IIA, and IIB. Total fiber number and area percentage were calculated, where fiber number percentage is the counted muscle fibers of each fiber type and fiber area percentage refers to the cross-sectional area of each fiber type divided by the total fiber area.

### 2.3. Meat Quality Measurement

The pH of the LT muscle was measured using a portable pH meter (Mettler Toledo, MP 230, Schwerzenbach, Switzerland). A total of 3 g of meat from the individual carcass was chopped and homogenized (IKA T25 ULTRA-TURRAX, Berlin, Germany) for 30 s in 27 mL of distilled water. The pH meter was first calibrated on 7, 4.01, and 9.21 before pH evaluation. The pH electrode was used for the determination of the pH of the resultant homogenates.

Color determination of the samples was carried out using a Konica Minolta Colorimeter (Chroma meter, CR-300, Tokyo, Japan) equipped with a standard D65 illuminant using a 2° position of the standard observer with a pulse xenon lamp and 8 mm reading surface area. L* (lightness), a* (redness), and b* (yellowness) values were recorded three times at three different locations. CIE a* b* values were used to calculate the saturation index ((a*2 + b*2)1/2) and hue angle ((b*/a*) tan − 1) according to AMSA guidelines [19]. The samples of the LT muscle were subjected to blooming at 25 °C for 30 min. Before each series of measurements, the instrument was calibrated using a white ceramic plate (Y = 93.5, X = 0.3132, y = 0.3198). The average value from three different locations from each sample group surface was used for statistical analysis.

Water-holding capacity (WHC) measurement of meat was carried out based on cooking loss and release water. Cooking loss (CL) percentage was determined as described by Hwang et al. [20]. Briefly, for CL%, the LT muscle was cut into 4 × 3 × 3 cm^3^ (L × H × W) dimensions with 25 g weight and packed in a low-density polyethylene bag. Samples were cooked at 75 °C for 30 min and chilled immediately in ice flakes for 10 min. CL% was determined by weight difference before and after cooking.

Release water (RW%) was determined following the method proposed by Joo [21]. A meat sample (3.0 ± 0.05 g) was placed on a previously dried and weighed filter paper (Whatman No. 1 of 11 cm diameter) with two thin plastic films. After weighing them, the filter paper and plastic film with meat samples were positioned between Plexiglas plates. A load of 2.5 kg and free mechanical force were applied for 5 min. Then, wet filter paper and plastic films were quickly weighed after carefully removing the compressed meat. The percentage of RW was calculated as follows: RW% = ((Wet filter paper and plastic film weight) − (dry filter paper and plastic film weight)/Meat sample weight) × 100%.

### 2.4. Tenderness-Related Measurements

Warner–Bratzler shear force (WBSF) was measured on cooked samples (from cooking loss) using the established procedure of AMSA [22]. A coring instrument (1.3 cm diameter) was inserted into cooked sample in the direction of the myofiber orientation. The cores were sheared perpendicular to the myofiber direction using an Instron tensile testing machine (Model 4443, Instron Corp., Boston, MA, USA) with a V-shaped shear blade. These sample cores were randomly collected from three different locations (caudal end, center, and cranial end) with less or no presence of connective tissue. Peak force was obtained using a 100-N load cell tension applied at a crosshead speed of 250 mm/min. The maximum peak force was reported as the shear force. From each sample, an average of five readings were taken.

Meat samples of 1 g were used to analyze sarcomere length (SL). Two borate-KCl buffer solutions—namely, solution A (0.1 M KCl, 0.039 M boric acid, and 5 mM EDTA in 2.5% glutaraldehyde) and solution B (0.25 M KCl, 0.29 M boric acid, and 5 mM EDTA in 2.5 glutaraldehyde)—were used as fixatives [23]. The raw meat was excised into small pieces (1 × 1 × 1 cm^3^) with the fibers running longitudinally and the core was placed in a capped vial with 20 mL solution A for 4 h. The sample was then transferred to a capped vial containing solution B and fixed overnight at 4 °C. Sample was removed from vial and the individual fibers were torn into fragments using tweezers. The fibers were spread apart (~10 μm) and placed on a clean microscope slide. A small drop of solution B was added to the slide and covered with a cover slip. Sarcomere length was determined by placing the slide straight into the path of a vertically oriented laser beam (Uniphase 1202-1, Manteca, CA, USA). Light diffraction patterns from these fibers were displayed on a screen. According to the following formula, sarcomere length (µm) was calculated:SL (µm) = (632.8 × 10^−3^ × D × [√(T/D)^2^ + 1])/T)
where D is the distance from the sample to diffraction pattern screen (D = 98 mm) and T is the separation (mm) between zero and the first maximum band.

The myofibrillar fragmentation index (MFI) was determined according to the method of Hopkins et al. [24] with minor changes by Nakyinsige et al. [25]. Briefly, 2.5 g of pulverized LT muscle was homogenized with 30 mL 20 mM ice-cold potassium phosphate buffer (pH 7.0) containing 100 mM KCl, 1 mM EDTA, and 1 mM MgCl_2_ (Wiggen Hauser, Berlin, Germany) for 60 s. The homogenates were then centrifuged at 1000× *g* for 15 min at 2 °C. The obtained supernatant was discarded while the pellets were re-suspended in 25 mL phosphate buffer and a centrifugation was repeated. Similarly, the pellets were suspended in 15 mL of buffer and vortexed after the resulting supernatant was discarded. In order to remove the remaining connective tissues, the myofibril suspensions were filtered into 50 mL conical centrifuge tubes using 1.0 mm polyethylene strainers and then rinsed with an additional 15 mL of buffer. The concentration of total protein for the final suspension was assessed by the procedure of Bradford [26]. Standards were prepared using bovine serum albumin and absorbance was measured at 595 nm using a spectrophotometer. The myofibril suspension was diluted with potassium phosphate buffer to a final protein concentration of 0.5 ± 0.05 mg/mL and the absorbance of the diluted myofibril suspensions were measured at 540 nm with a UV-Vis spectroscopy system (Agilent 8453, Santa Clara, CA, USA). Triplicate absorbance readings were averaged and multiplied by 150 to obtain the index values for myofibrillar fragmentation.

### 2.5. Histology Features of Perimysium

The perimysium thicknesses (PMT) of raw LT muscle was set and projected based on the method Flint and Pickering [27] and modified by Li et al. [28]. Samples of 0.5 × 0.5 × 0.5 cm^3^ were excised from LT muscle and rapidly frozen in liquid nitrogen. Transverse serials of 10 µm were cut perpendicularly to the direction of muscle fibers using a cryostat microtome (Microm GmbH, HM525, Walldrof, Germany) at −25 °C. Samples were first fixed in acetone at 0 °C for 12 h to minimize cytoplasmic staining, then subsequently transferred into Bouin’s solution (75 mL picric acid, 25 mL 10% formalin, and 5 mL glacial acetic acid), and placed in a cold environment for 30 min. These samples were then stained in saturated picro-sirius red solutions (C.I 35782; direct red 80, Sigma. Buchs, Switzerland) for 1 h in a dark environment. After staining, samples were washed in 0.01 M HCl solution for 2 min and dehydrated in 100% ethanol for 5 min. Finally, they were submerged in xylene for 2 min and covered with Entellan mounting medium and a glass coverslip. Images were examined using a light microscope (BX41, Olympus, Tokyo, Japan). Primary and secondary perimysia thicknesses at five different points on five images were obtained with Image-Pro Plus (Image-Pro^®^plus 5.1, Media Cybernetics Inc., Rockville, MD, USA). A total of 25 measurement points in the area were randomly selected to obtain the mean value of primary and secondary perimysia thicknesses.

### 2.6. Statistical Analysis

A two-way analysis of variance (ANOVA) was performed in order to evaluate the statistical significance (*p* < 0.05) of the effect of slaughter age on meat. Analysis of the main effect of each independent variable and any significant differences between them was done using Duncan’s multiple range test, performed as a function of animal age and storage days. Animal age and storage time were assigned as fixed effects, and the replication was considered as a random effect. Complete randomized design was adopted for statistical analysis. The error terms used throughout this study are standard error (SE). Results are expressed as least square mean values of three independent replications, except for WBSF, for which the average of five measurements was obtained using repeated measurements. Pearson correlation coefficient was determined in order to describe the relationship between meat quality traits and muscle fiber characteristics using SAS PROC.

## 3. Results and Discussion

### 3.1. Histochemical Characteristics

Muscle fibers in LT muscles obtained from KNBG were divided into type I, IIA, and IIB, as shown in Figure 1. The results of muscle fiber number (%) and muscle fiber area (%) of LT muscle from AG and YG are presented in Figure 2. Numerically counted fiber number percentages of type I were significantly higher in AG as compared to YG (*p* < 0.05). In contrast, to type I fiber, fiber number percentages of type IIA fiber were higher in YG than those in AG (*p* < 0.05). The results of muscle fiber area showed a similar pattern to muscle fiber number.

Many factors are known to contribute to fiber type variation, such as sex [29], age [30], and breed [31]. Picard et al. [32] have shown that the percentage of type I fibers is increased significantly in male Montbeliard cattle between 66 and 170 d of age. Similarly, in meat animals and poultry, Swatland [33] mentioned that muscle fibers undergo a continual alteration throughout life as an adaptation to changing functional demands and that “fiber type” merely reflects the constitution of fiber at any particular time. Type IIB fibers were the same among AGs and YGs in the present study, similar to the description of Wegner et al. [34] in breed and growth variations in cattle. Current results clearly show that the percentage of type I fiber increased and the percentage of type IIA fiber decreased as the goats aged from 9 to 18 months.

### 3.2. Muscle pH, Meat Color, and WHC

Changes in meat quality traits including pH, color, and WHC measurements are shown in Table 1. There were no significant differences in pH between AGs and YGs during 14 days of cold storage (*p* > 0.05). However, at day 21, AGs had higher pH values than YGs (*p* < 0.05). The sudden variation in pH could be due to different metabolism in different animals. Similarly, Mach et al. [35] reported the relationship between carcass conformation and meat pH in beef. They also proved that the increase of fatness resulted in a considerable decline in the frequency of meat with high pH. An increase in age also leads to increased goat meat pH, according to Marichal et al. [36].

Post-mortem storage time had a significant influence on L* values, which increased significantly as storage days advanced (*p* < 0.05), while a* values showed an increasing and decreasing pattern during post-mortem storage days. However, b* values decreased during post-mortem storage. Regarding the goat age, there was no significant difference in lightness value (L*) between AGs and YGs, although redness (a*) values on days 0 and 7 were significantly higher in AGs than those in YGs. The high a* value of AGs could be attributed to the high percentage of type I fibers in AG muscle. As shown in Table 2, redness values of goat meat showed a positive correlation with type I fiber number % (*r* = 0.66) but a negative correlation with fiber number of type IIB (*r* = −0.67). Additionally, the correlation coefficient between redness values and percentage of muscle fiber types was similar to Hwang et al. [20], who demonstrated in Korean native steers that a high number of type I fibers and a low number of type IIB fibers led to increased intensity of redness values.

Increasing and decreasing trends in L*, a*, and b* values were noted throughout post-mortem storage. The improvement in L* and a* over storage agrees with previous findings in chevon [37] and beef [38]. Additionally, Ilavarasan et al. [39] examined similar findings for lightness and redness values in young and adult goat meat. However, there was no significant difference in yellowness values between AGs and YGs.

Slaughter age and post-mortem time had no influence (*p* > 0.05) on CL% between AGs and YGs—a stable configuration was observed throughout the experimental trial. Nevertheless, the RW% of YGs was significantly higher than that of AGs, and it increased as storage continued both in AGs and YGs (*p* < 0.05). De Palo et al. [40] also stated that horse slaughter at various ages (6, 11, and 18 months) had no significant effect on CL%. The higher RW% of YG could be attributed to the low percentage of type I fiber and the high percentage of type IIA fiber in YG muscle (Figure 2). Furthermore, in the current study, RW% showed a negative (*p* = 0.1067) correlation with type I but a positive correlation with type IIA and IIB fibers (Table 2). Our previous studies also indicated the positive correlation between the percentage of white muscle fibers (type IIA and IIB fibers) and drip loss (%) in pig [41] and cattle [20].

### 3.3. Tenderness

Changes in tenderness-related measurements including WBSF, SL, and MFI during 21 days of cold storage are shown in Figure 3. A decreasing trend in WBSF values was seen from 0 to 21 days of post-mortem storage (*p* < 0.05; Figure 3A). Additionally, AGs had significantly higher WBSF values than YGs (*p* < 0.05). SL increased with advancing storage period from 0 to 21 days post mortem (*p* < 0.05; Figure 3B). The age effect on SL had no clear difference between AGs and YGs. However, at day 7 of post-mortem storage, YGs had higher SL than AGs (*p* < 0.05). MFI increased significantly with advancing post-mortem storage time from 0 to 21 days, as shown in Figure 3C (*p* < 0.05). Regardless of post-mortem time, YGs had significantly higher MFI than AGs (*p* < 0.05).

Current observations suggest that young slaughter age (9 months) and 21 days of post-mortem time are key factors leading to the tenderness of goat meat. During storage, WBSF values decreased as post-mortem time advanced. The decline in the WBSF value was due to endogenous muscle proteinases that can weaken the myofibrillar structure. Post-mortem tenderization of meat is mainly derived from the changes caused in the muscle during post-mortem aging [42]. On the other hand, in terms of age, the WBSF values of AGs were higher than those of YGs. This was probably because older animals have more crosslinks of connective tissues than younger animals. This crosslinking of connective tissue mainly contributes to the background of toughness in goat meat [43]. Therefore, the current results of WBSF values were also supported by the present PMT findings, which showed that AGs had higher PMT as compared to YGs, making YG meat tenderer than AG meat. Consequently, our data suggest that AG loin is tougher than YG loin regardless of its aging time because of its thick perimysium.

However, based on AMSA [44] regulation, the optimum meat storage times are different for different species—10 to 28 days for beef, 5 to 14 days for pork, and 7 to 14 days for lamb. In the current study, 0 to 21 days of storage were tested to determine the optimum storage time for goat meat in adult and young goats. Previously, Shackelford et al. [45] categorized WBSF values in beef into three tenderness groups based on their threshold levels: tender (<3.9 kg), intermediate (3.9–4.6 kg), and tough (>4.6 kg). The variations in WBSF values with standard threshold levels could be due to different species, breed, and nutrition of the animals. Calkins et al. [46] observed correlations in cattle among fiber diameter, fiber type composition, and meat shear force values, although the precise relationship remains undefined. According to Silva et al. [47], younger Nellore animals (slaughtered after 0 days on feed) had higher post-mortem meat tenderization. Although these results are variable and sometimes contradictory, the current observations suggest that relationships exit between muscle fiber characteristics and meat quality, particularly in goat. A strong positive correlation was found between WBSF and muscle fiber area for type I and type IIB fibers (*r* = 0.69 and *r* = 0.76, respectively), while a strong negative correlation was observed between WBSF and muscle fiber area of type IIA (*r* = −0.71; Table 2). Our data indicate that YG loin is tender because it has a higher percentage of type IIA fiber than AG loin.

As expected, SL and MFI increased dramatically during the storage period while WBSF decreased for 21 days of post-mortem storage. Longer sarcomeres in beef samples are associated with better tenderness [48]. The present finding was in line with that of Pen [49], who observed in beef that the rise in SL was linked to an increase in post-mortem time. In bovine muscle fibers, longer sarcomeres are more liable to post-mortem proteolysis [50]. Our data show that goat age did not affect SL but has affected MFI. YGs had significantly higher MFI than AGs during 21 days of cold storage. This might be one of the reasons why YG is tenderer than AG. However, our results deviate from those of Ilavarasan et al. [39], who reported that there was a highly significant increase in MFI with higher values in adult goat meat as compared to that of young goats. The current findings prove an association with type I fiber and MFI in LT muscle, as a negative correlation was observed with both type I fiber number and fiber area type I (*r* = −0.73 and *r* = −0.80, respectively; Table 2).

### 3.4. Perimysium Thickness

The PMT of AG was significantly higher than that of YG (*p* < 0.05), while days post mortem had no effect on PMT as shown in Figure 4 and Figure 5. Goat age had a significant impact on PMT, with very thick perimysium in AGs as compared to YGs. Previously, Wojtysiak [51] described that pork slaughtered at different ages (90, 150, and 210 days) had significant differences in PMT (22.16, 31.67, and 41.53 µm, respectively). These structural arrangements are similar to those noted earlier in the skeletal muscle of various animal species.

In general, meat toughness is increased with increasing PMT in pig [52] and cattle [53], which are consistent with the present findings in goat. Similarly, Roy et al. [54] found that the difference between the thicknesses of the primary thick perimysium of the tough and tender muscles approached significance (*p* = 0.166), as did that of the secondary thick perimysium (*p* = 0.058) in tender muscles compared with those that were tough. Moreover, adipocytes may play a major role in increasing the thickness of the perimysium by occupying space and expanding collagen fiber bundles in horse meat. An et al. [55] have also described differences in PMT in broiler chicken at 6 weeks of age and White Leghorn chicken at 6 and 18 weeks of age. The PMT of broilers was much smaller than that of White Leghorns. The elder White Leghorns had smaller epimysium thickness but greater perimysium thickness than broilers at 6 weeks of age. Astruc [14] described that perimysial collagen was degraded during storage. The major post-mortem change in perimysium is that myofibers separate from perimysium within 6 h. Perimysium seems to be the structure that is most vulnerable to meat shearing action. It has also been shown that the isometric tension of intramuscular collagen decreases at 21 days post-mortem in beef. Furthermore, the breaking strength of the perimysial connective tissue in raw beef decreases during post-mortem days, although their results deviate from our finding. In the present study, PMT had strongly positive correlations with fiber area of type I (*r* = 0.80) while it had negative correlation with fiber area of types IIA and IIB (*r* = −0.82 and *r* = −0.87, respectively). Based on the current data, it is suggested that the PMT of AGs is thicker than that of YGs due to its high percentage of type I fibers and low percentage of type IIA fibers.

## 4. Conclusions

Muscle fiber composition in the LT muscle of goat meat depends on age, and as a result, the meat quality traits such as color, release water, and tenderness also vary. The percentage of fiber types I and II increased and decreased at slaughter ages 9 and 18 months, respectively. As a result, AG meat had a high a* value, low RW%, and high WBSF value compared to YG meat. In particular, AGs had less tender meat characteristics than YGs due to thicker perimysium as well as muscle fiber characteristics, which was not overcome by 21 days of storage. Regardless of age, the storage period had a great effect on meat quality traits, especially tenderness.

## Figures and Tables

**Figure 1 foods-08-00571-f001:**
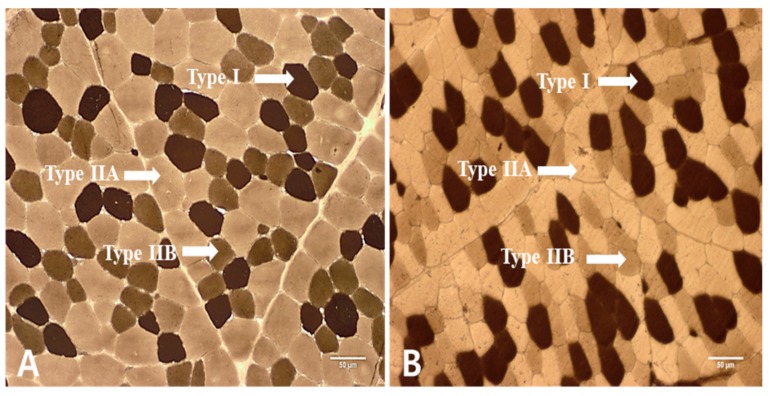
Sections of *longissimus thoracis* (LT) muscle. Stained for myosin ATPase reactivity after preincubation at pH 4.50. Magnification of 100× was used (Bar = 50 µm). (A: AG: adult goat (18 months); B: YG: young goat (9 months)).

**Figure 2 foods-08-00571-f002:**
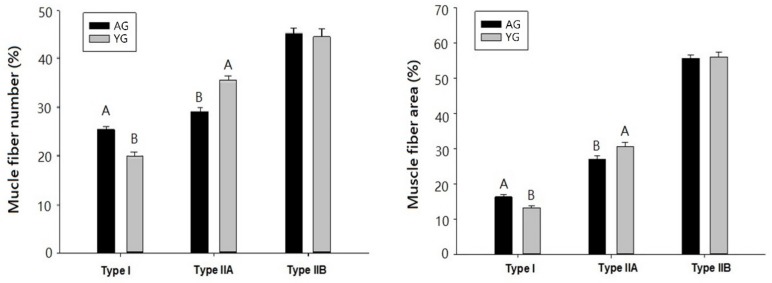
Muscle fiber number and area composition of goat LT muscle. Error bars represent standard error. Different letters on the bars denote significant differences between slaughter ages (*p* < 0.05).

**Figure 3 foods-08-00571-f003:**
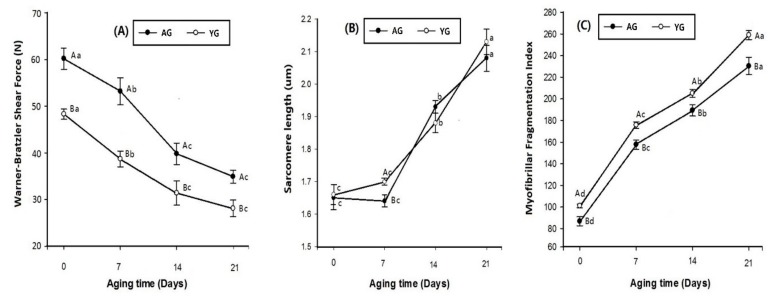
Warner–Bratzler shear force (**A**), sarcomere length (**B**), and myofibrillar fragmentation index (**C**) of goat meat LT muscle. Different letters in the figure indicate significant differences for days post mortem (a–c) and slaughter age (A–B) (*p* < 0.05).

**Figure 4 foods-08-00571-f004:**
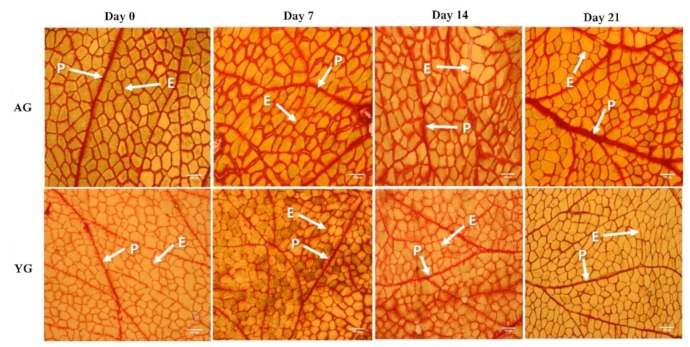
Thickness of perimysium during post-mortem storage days at different ages of goat LT muscle). P = perimysium; E = endomysium; Bar = 200 μm.

**Figure 5 foods-08-00571-f005:**
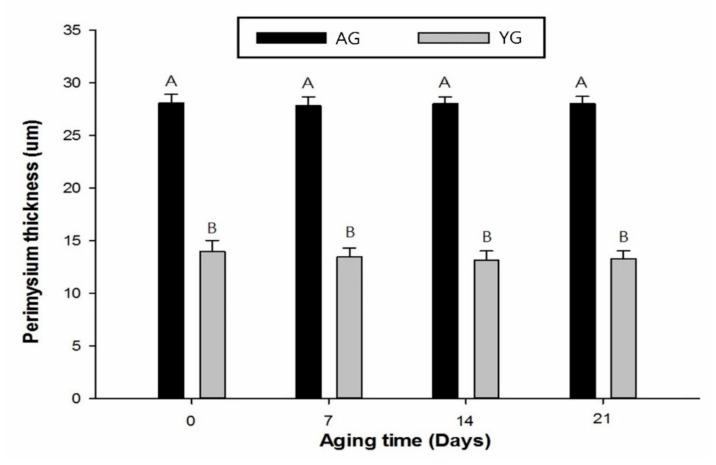
Perimysium thickness during post-mortem storage of goat meat slaughtered at various ages. Error bars represents mean ± standard error. Different letters on the bars denote significant difference (*p* < 0.05).

**Table 1 foods-08-00571-t001:** Meat quality traits of goat meat LT muscle of different ages during cold storage.

Parameter	Age	Storage Days			
		0	7	14	21
pH	AG	5.62 ± 0.02	5.63 ± 0.02	5.64 ± 0.03	5.67 ± 0.02 ^A^
	YG	5.60 ± 0.01	5.59 ± 0.03	5.61 ± 0.02	5.58 ± 0.01 ^B^
CIE L*	AG	42.18 ± 0.41 ^b^	44.56 ± 0.38 ^a^	44.28 ± 0.70 ^a^	45.28 ± 0.50 ^a^
	YG	42.48 ± 0.50 ^b^	44.78 ± 0.56 ^a^	44.88 ± 0.36 ^a^	45.40 ± 0.27 ^a^
CIE a*	AG	21.12 ± 0.16 ^Ab^	22.02 ± 0.12 ^Aa^	21.08 ± 0.20 ^b^	20.54 ± 0.71 ^c^
	YG	20.70 ± 0.06 ^Bb^	21.44 ± 0.15 ^Ba^	20.62 ± 0.06 ^b^	20.40 ± 0.08 ^c^
CIE b*	AG	5.22 ± 0.20 ^a^	4.31 ± 0.12 ^b^	4.36 ± 0.05 ^b^	4.47 ± 0.11 ^b^
	YG	4.76 ± 0.20 ^a^	4.36 ± 0.20 ^b^	4.31 ± 0.21 ^b^	4.17 ± 0.13 ^b^
Cooking loss (%)	AG	18.16 ± 0.47	17.96 ± 0.21	17.88 ± 0.62	17.78 ± 0.76
	YG	18.40 ± 0.50	18.22 ± 0.16	17.70 ± 0.28	17.74 ± 0.38
Released water (%)	AG	5.72 ± 0.13 ^Bb^	6.05 ± 0.19 ^Bab^	6.41 ± 0.20 ^Ba^	6.01 ± 0.14 ^Bab^
	YG	6.50 ± 0.50 ^Ab^	6.99 ± 0.35 ^Aab^	7.37 ± 0.24 ^Aa^	7.20 ± 0.14 ^Aab^

Different superscript letters in the same row (a–c) for different post-mortem days and in the same column (A–B) for different ages indicate that means are significantly different (*p* < 0.05).

**Table 2 foods-08-00571-t002:** Correlation coefficients (r) between histochemical characteristics and meat quality traits of goat LT muscle.

Measurements	Fiber Number Percentage			Fiber Area Percentage		
	Type I	Type IIA	Type IIB	Type I	Type IIA	Type IIB
PMT	0.56	−0.40	−0.63	0.80 **	−0.82 **	−0.87 **
WBSF	−0.47	−0.49	0.58	0.69*	−0.71 *	0.76 *
SL	−0.15	0.09	0.15	−0.04	0.15	−0.02
MFI	−0.73 *	0.18	0.63	−0.80 **	0.48	0.74 *
RW%	−0.37	0.41	0.47	−0.59	0.67 *	0.67 *
CL%	0.19	0.32	0.29	−0.06	0.01	0.03
L*	0.42	0.57	−058	0.34	−0.01	0.23
a*	0.66 *	−0.38	−0.67 *	0.73 *	−0.53	−0.70 *
b*	−0.15	−023	0.22	0.33	−0.46	0.40

* *p* < 0.05, ** *p* < 0.01. PMT: perimysium thickness; WBSF: Warner–Bratzler shear force; SL: sarcomere length; MFI: myofibrillar fragmentation index; RW%: released water; CL%: cooking loss; L*: lightness; a*: redness; b*: yellowness.

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
