# Peer review of "Effect of Slaughter Age on Muscle Fiber Composition, Intramuscular Connective Tissue, and Tenderness of Goat Meat during Post-Mortem Time"

_foods, 2019, doi:10.3390/foods8110571_

Round 1

Reviewer 1 Report

The article presents studies assessing the influence of age of goats on meat quality and muscle fibre profile of meat.

General remarks:

The article is written in the correct English language but has serious methodological flaws:

The authors did not provide in the description of the methodology what was the sex of goats from which meat samples were taken for testing yes in relation to 9 and 18 months old animals. The description states that 24 number of LD muscles were used in the study, whereas 6 animals at the age of 9 months and 6 animals at the age of 18 months. There is a contradiction in the information. There are no cooking or sample handling conditions in the methodology before WBSF measurements. A major drawback of the work is the lack of analysis of the chemical composition of meat samples: dry matter, crude fat, total protein, ash and water content. Chemical composition of meat and mutual relations between its components are one of the most important factors influencing tenderness and shear force.

It is difficult to evaluate the results without first mentioning the sex of the animals and explaining the number of animals and samples that have been analysed.

Reviewer 2 Report

The manuscript presents interesting data regarding goat meat quality traits as affected by age at slaughter and postmortem storage time. The experimental control is questionable, since the samples were collected in a commercial slaughterhouse (I mean, we don’t know much about these animals’ background in terms of nutrition and management). The methods used are well described, however, it is still lacking some details regarding the statistical analysis. The text is understandable, but there are some sentences which the wording could be improved. Although I am not a native English speaker, I attempted to suggest some modifications.

General comments and questions:

Do you have any background information of these animals? Were they from the same farm? Did they have the same genetic background? Were they slaughtered in the same day, under the same conditions? You collected 24 LD (it is understandable considering the LD size of goat), but did you consider it as 24 independent samples or 12 independent samples? When you say 1, 7, 14, and 21 days, are you referring to postmortem day? I mean, the sample 1 was frozen 24h postmortem or 48h postmortem? I think you should consider expressing the aging as “days postmortem”. Did you evaluate the PMT in the LD or semitendinosus? If it was in the semitendinosus you should describe it in the “samples” subsection and whenever you discuss the PMT data. Did you evaluate PMT in Raw and cooked meat as described in the L. 127-128? Where are these data? Is it possible to measure the endomysium thickness? Could it help in the discussion? Is it possible to evaluate fiber size (diameter)? The Statistical Analysis section should be improved. Please provide the statistical model used. How were the factors animal age and postmortem time analyzed? I mean was used repeated measurements or a factorial design? Did you analyze all 24 LD individually or the 12 goats? Please add a sentence describing how the correlations were calculated. Which SAS PROC was used?

Specific comments and suggestions:

Tittle: I think the term “during postmortem aging” is redundant, since aging refers to the postmortem time. Please review and consider changing it. This should be checked in the body of the manuscript too (e.g., L. 12).

Introduction

You should add some sentences contextualizing the importance of the perimysium on meat tenderness.

L.13 – Delete “features of”

L.15 – Were all the 24 LD analyzed? This sentence gives us the idea that you had 24 animals when you had 12.

L.15 – The sentence “from 9 months young goat (YG)” should be reviewed.

L.22 – I suggest deleting “even if it ages 21 days”.

L.28-29 – Please provide examples of nutritional values which are better in the goat meat compared to the others.

L.30 – Please define “KNBG”

L.47 – I didn’t understand what you want to say with “on meat cuts”.

L.48-49 – Please explain better this idea.

Materials and Methods

L.55 – Again please correct the “9 months young goat”.

L.56-59 – This should be the first sentence of the Material and Methods section.

L.60-61 – You divided the LD into 5 steaks, but you have 4 postmortem times (1, 7, 14, and 21). What not divide into 4? Also, please explain better how the steaks were assigned to the postmortem times.

L.74-76 – Please rewrite these sentences to make it more comprehensive.

L.84 – You probably want to say “color” instead of “calorimetric”.

L.89 – Do not use “%” alone.

L.89 – Delete “formerly”

L.90 – I suggest replacing “was based on the method of” by “was performed following the method proposed by”.

L.92 – Please replace “electrical balance” by “scale”

L.95 – Replace “damp” by “wet”

L.96 – Add “dry” before “filter-paper”

L.101 – Add “the” before established

L.102 – Please rewrite “Toward parallel into myofiber direction”

L.105 – Replace “gained” by “obtained by”

L.105-106 – What does “The full-scale load was 50 kg” mean?

L.117 – Please replace “µ=” by “SL (µm)=”

L.27 – Was this muscle used?

L.129-130 – Delete “for 6h”.

L.148 – Shouldn’t the “p” be capitalized? Please double check it in the whole manuscript, including tables and figures.

L.148-149 – I couldn’t understand this sentence. Can you explain better?

Results and Discussion

L.158 – I suggest replacing “tendency” by “pattern”. The term “tendency” is usually used to refer to statistical significance.

Figure 1. – Can you adjust the contrast of the pictures? For example, the fiber type IIA in the picture B looks more like type I of the figure A. So, I think if you change a little bit the contrast of one picture will be easer for the reader recognize the fiber types.

Figure 1. – Is the fiber type showed correctly? I have the impression that the type I and type IIB are exchanged, based on the figure (I mean, looking at the figure one, apparently, we have more type I than type IIB, opposite of what is showing in the figure 2).

L.160 – delete “Serial”. Also, please provide the animal age in all figure captions and table labels.

L.163 – Delete “formerly”

L.172 – Please add “between slaughter ages” after “differences”.

L.177-178 – What could be affected the meat pH during the postmortem time? Please explain it better.

L.184-185 – Please delete “Changes in color measurements (L*, a* and b*) of OG and YG during postmortem aging time are shown in Table 1.” It is the same information in the L.177-178.

L.185-186 – Please double check the sentence. It is not true for a*.

L.199 – Why we could have a decrease in myoglobin concentration?

L.203-206 – Please move these sentences to the previous paragraph (L.191).

L.205 – Replace “type 1” by “type I”

L.207 – I suggest avoiding use the term “trend” in this context. Please replace it by another word.

L.212-213 – It seems the correlation between RW and type I fiber was not significant. So, please provide the P-value in the text.

L.218 - I suggest avoiding use the term “trend” in this context. Please replace it by another word.

L.219 – I suggest replacing “(P < 0.05) (Figure 3A).” by “(P < 0.05; Figure 3A).” Please check this across the whole manuscript (e.g., L.221).

L.224 – I think you want to say aging time instead of “age of goat”. Please check this information.

L.227 – There is no line in the figure 3. So, Please check the term “in line”.

L.229 – Please change the wording of this sentence “The present results show that tenderness of goat meat is crucial to age and aging period”. I couldn’t understand what you want to say here.

L.232 – I suggest replacing “state” by “condition”.

L.229-239 – I think you should discuss better about meat tenderness here, since the goal of postmortem storage is to improve tenderness. So, tell the readers what is considered a tender meat (I mean, what is the threshold in terms of WBSF to consider a certain meat tender). I then how long should we store the meat from young and old goats to get at this threshold? Although the collagen is one of the most determinants in making old animals’ meat tough, some studies have been shown that young animals has higher postmortem tenderization rate, as you found here. Please, see this recently published paper (https://doi.org/10.1016/j.livsci.2019.09.012) showing that young animals had higher meat tenderization. Consider adding it to your discussion.

L.238 – Pleaser replace “more tenderer” by “more tender”

L.240-241 – Please rephrase this sentence.

L.251 – I suggest replacing “Legthier” by “Longer”.

L.255 – Please delete “activity”

L.250-254 – I am not sure why the SL increased during the postmortem storage. In my opinion, it shouldn’t change after rigor mortis. In our lab when we evaluated it after 24h postmortem there was no difference. Also, as you can see on the paper of King et al. (2019) [doi:10.22175/mmb2018.09.0027], there was no difference on SL across the postmortem time. Although the change was not big in your case (i.e., 0.4um), I still thing what could cause this. Do you have any idea?

L.258-259 – Please check the wording of this sentence.

L.262-263 – Were you expecting a change on PMT postmortem? Please, discuss this.

L.264 – Replace “.Previously,” by “. Previously”

L.276-280 – Please check the wording of this phase. It is too long and hard to follow the idea.

L.284 – I think the correlation was “positive” not “negative”. Please double check it.

L.285 - I think the correlation was “negative” not “positive”. Please double check it.

Conclusion

L.290 – Do not use abbreviation here. Please replace “WHC” by the appropriate analysis.

L.291 – Please check the wording in the sentence “The percentage of type I fiber increases and the percentage of type IIA fiber decreases during the age of goat from 9 months to 18 months”

L.293 – I suggest replacing “tougher characteristics” by “less tender meat”

L.294-295 – Please double check the wording in this sentence. I think we have too many “the”.

Round 2

Reviewer 1 Report

The revised version of the article was written taking into account the comments contained in the review to the first version. Comments on the description of parts of the methodology, including statistics, have been taken into account. The chapter on results and discussion has also been completed, with no reservations.

Author Response

We wish to thank you all for your constructive comments in current review. Your comments provided valuable insights to refine its contents and analysis. In this manuscript, we try to address the issues raised as best as possible.

Reviewer 2 Report

The authors have made some improvements in the manuscript. However, a lot of my questions and suggestions were not taken in account or were partially answered.

The statistical analysis needs to be profoundly reviewed. Please see the questions below.

Although the manuscript is understandable, I highly recommend a language checking before publication. For instance, the authors edited many sentences (e.g., L. 15; L. 65; L.82-84; L. 111; L. 224; L.235; L.248-249; L.285-286; L.304-307), but these sentences still need improvements.

Unanswered question and suggestion:

Please add to the manuscript a brief description of the animals’ background, describing how was the feed management and where the animals were raised. Again, it is not clear for me how did you managed the meat samples. When you say 1, 7, 14, and 21 days, are you referring to postmortem day? I mean, the sample 1 was frozen 24h postmortem or 48h postmortem? For instance, in the L. 67-69 you say “Approximately after 24 h of slaughtering, the LD muscles were excised from right and left sides of carcasses, weighed and divided equally into four parts, and 5 cm thickness steaks were prepared, vacuum packed, and stored at 4°C for 1,7, 14, and 21 days”. So, I understand that the meat has stored for 21 days after deboning, which would be 22 days postmortem not 21 day postmortem. So, please confirm this information. If the endomysium thickness is important, I think you should evaluate it. I think you should evaluate fiber size (diameter). The Statistical Analysis section should be improved. Please provide the statistical model used (I mean, was it used a Latin square, complete randomized design, block design?). How were the factors animal age and postmortem time analyzed? I think you should use for the analysis an average per animal not the replicates (note, there is biological replication and technical replication). Please add a sentence describing how the correlations were calculated. Which procedure of SAS was used? The statistical section is unsatisfactory. Please make changes describing in detail the analysis performed. Please, be consistent showing all the “p” of the p-values in lowercase. Check it throughout the manuscript.

Specific comments and suggestions:

L.28-29 – Please provide examples of nutritional values which are better in the goat meat compared to the others. Add the information to the introduction, supporting why KNBG meat is healthy.

L.67-68 - Please explain better how the steaks were assigned to the postmortem times. It is still not clear.

L.135-136 – Please double check this sentence. I my understanding you are using the semitendinosus muscles.

Figure 1. – Can you adjust the contrast of the pictures? For example, the fiber type IIA in the picture B looks more like type I of the figure A. So, I think if you change a little bit the contrast of ONE picture will be easier for the reader recognize the fiber types. Please do this change if possible.

Figure 1. – Is the fiber type showed correctly? I have the impression that the type I and type IIB are exchanged, based on the figure (I mean, looking at the figure one, apparently, we have more type I than type IIB, opposite of what is showing in the figure 2). Please, see how the fiber types were assigned in the Hwang et al. (2017) [https://doi.org/10.5851/kosfa.2017.37.6.948] paper.

L.192-196 – This is truth for the first 24 h but not for 21 days. Please, explain how meat pH can change after the 24 h postmortem.

L.229 – Please provide the exact p-value.

L.268-270 – You didn’t evaluate sex condition. The paper of Silva et al. (2019) shows that younger animals (slaughtered after 0 days on feed) had higher postmortem meat tenderization. Please, edit the discussion or consider removing this statement.  
